# Detection of Diabetic Retinopathy in Retinal Fundus Images Using CNN Classification Models

**Al-Omaisi Asia** [1], **Cheng-Zhang Zhu** [1,2,*], **Sara A. Althubiti** [3], **Dalal Al-Alimi** [4], **Ya-Long Xiao** [1,2], **Ping-Bo Ouyang** [5] **and Mohammed A. A. Al-Qaness** [6,7]

1  School of Computer Science and Engineering, Central South University, Changsha 410083, China
2  College of Literature and Journalism, Central South University, Changsha 410083, China
3  Department of Computer Science, College of Computer and Information Sciences, Majmaah University, Al-Majmaah 11952, Saudi Arabia
4  School of Computer Science, China University of Geosciences, Wuhan 430074, China
5  Department of Ophthalmology, The Second Xiangya Hospital of Central South University, Changsha 410083, China
6  State Key Laboratory for Information Engineering in Surveying, Mapping and Remote Sensing, Wuhan University, Wuhan 430079, China
7  Faculty of Engineering, Sana'a University, Sana'a 12544, Yemen
*  Correspondence: chzhzhu@csu.edu.cn

**Abstract:** Diabetes is a widespread disease in the world and can lead to diabetic retinopathy, macular edema, and other obvious microvascular complications in the retina of the human eye. This study attempts to detect diabetic retinopathy (DR), which has been the main reason behind the blindness of people in the last decade. Timely or early treatment is necessary to prevent some DR complications and control blood glucose. DR is very difficult to detect in time-consuming manual diagnosis because of its diversity and complexity. This work utilizes a deep learning application, a convolutional neural network (CNN), in fundus photography to distinguish the stages of DR. The images dataset in this study is obtained from Xiangya No. 2 Hospital Ophthalmology (XHO), Changsha, China, which is very large, little and the labels are unbalanced. Thus, this study first solves the problem of the existing dataset by proposing a method that uses preprocessing, regularization, and augmentation steps to increase and prepare the image dataset of XHO for training and improve performance. Then, it takes the advantages of the power of CNN with different residual neural network (ResNet) structures, namely, ResNet-101, ResNet-50, and VggNet-16, to detect DR on XHO datasets. ResNet-101 achieved the maximum level of accuracy, 0.9888, with a training loss of 0.3499 and a testing loss of 0.9882. ResNet-101 is then assessed on 1787 photos from the HRF, STARE, DIARETDB0, and XHO databases, achieving an average accuracy of 0.97, which is greater than prior efforts. Results prove that the CNN model (ResNet-101) has better accuracy than ResNet-50 and VggNet-16 in DR image classification.

**Keywords:** classification; diabetic retinopathy; deep learning; CNN; ResNet; VggNet

## 1. Introduction

Diabetic retinopathy (DR) is a microvascular entanglement of type 1 and 2 diabetes. DR causes retinal irregularities and is one of the main causes of visual impairment around the world. Approximately 33% of individuals with diabetes have DR, and practically all diabetics will foster it after some time. By 2030, DR is predicted to afflict 191 million individuals [1,2]. Though the visual weakness and deficiency brought about by DR can be prevented [3], early discovery is critical [4]. To guarantee early discovery and brief treatment, current rules propose that individuals with inadequately controlled diabetes ought to be evaluated for DR once a year. Patients previously determined to have DR ought to be checked frequently. Evaluating for DR mainly includes catching a fundus picture of the retina, which is then assessed by expert ophthalmologists. If not treated, diabetic

retinopathy progresses through four clinical stages of mild nonproliferative, moderate nonproliferative, and severe nonproliferative, as shown in Figure 1.

**Figure 1.** Stages of diabetic retinopathy.

The number of retina photographs generated via the screening program will rise with the number of diabetic patients, challenging the provision of specialized eye care for everyone, thereby imposing a huge labor-intensive burden on medical experts and costs for medical services; these issues and the growing waiting list for ophthalmic consultations are the main problems facing public health systems [5–7]. These problems can be alleviated by an automated system, which can be used as a support tool for medical experts or a complete diagnostic tool. Many studies have reported on the application of deep learning (DL) algorithms in the automatic detection of DR. These techniques demonstrate the great sensitivity and specificity of automatic detection systems based on deep learning artificial neural networks to the reference DR. In addition, new eye problems such as age-related macular degeneration, glaucoma, and diabetic macular edema have recently been discovered [8–11].

The automated system must be able to arrange retinal photographs in accordance with the severity of clinical practice, such as the suggested international clinical levels of diabetic macular edema and DR, which are also used in some countries, for it to be practically practicable. According to previous studies, the latest experimental results of the former DR scale can be obtained, but the latter is not used for the experimental classification of maculopathy. The enormous amount of annotated images needed for model learning is a significant barrier to the widespread and successful usage of deep learning systems [12–14].

This study aims to use fundus image classification technology to automatically diagnose DR, classify fundus images according to the severity of DR, and realize end-to-end real-time classification from fundus photographs to the patient's condition. The automatic diagnosis system has a high degree of automation and precision, thereby reducing the pressure on DR diagnosis and treatment. This task uses different image preprocessing methods to extract numerous significant features and then classifies them. Therefore, this work uses three types of CNN architecture (residual neural network (ResNet)-101, ResNet-50, and VggNet-16) to detect the DR of two datasets and evaluates the training accuracy, training loss, and test accuracy of the model.

This paper introduces a CNN method based on deep learning to solve the problem of DR classification. The motivations for employing different CNN-based deep neural networks evolved and obtained a substantial result on ImageNet Challenger, the most important image classification and segmentation challenge in the image-analyzing area. In medical categorization, a CNN-based deep neural system is commonly utilized. Because CNN is such a good feature extractor, using it to identify medical pictures may save time and money on feature engineering. The key benefits of CNN are a low reliance on preprocessing and a reduction in the amount of human labor required to create its features. It is simple to comprehend and apply. Among all picture prediction algorithms, it has the greatest accuracy. It finds the relevant traits without the need for human intervention. In addition, CNN is computationally efficient. Finally, CNN has shown to be quite effective in resolving picture categorization issues. Many image databases' top performance has been greatly enhanced thanks to research based on CNN.

To the best of our knowledge, this is the first article to discuss the two categories of DR (symptomatic and asymptomatic) using different kinds of CNN architecture (ResNet-101, ResNet-50, and VggNet-16) on our laboratory dataset (obtained from the Xiangya Second Ophthalmic Hospital, Changsha, China, from March 2016 to October 2016). Recently, several new methods were introduced to adapt the CNN to large datasets. Then, we analyzed the performance and function of the different kinds of CNN architecture and used deep CNNs (ResNet-101, ResNet-50, and VggNet-16) to detect DR, ultimately achieving automatic and precise detection to minimize optical impairment. Compared with prior techniques, this work applied enhancements in terms of the convergence time of large-scale experimental datasets and classification performance. The main contributions of this paper are as follows:

- This work's laboratory database has a very large size, is little, and has some noise, and deep networks are slightly difficult to train. Thus, this study uses some preprocessing methodologies on the laboratory datasets to increase the training dataset and improve the performance of using a deep training network, such as resizing of input images, augmentation, and data normalization.
- This study demonstrates that CNN-based approaches could achieve state-of-the-art performance in DR detection. In addition, this work uses CNN methods to find the fundamental picture features relevant to disease grade prediction. Later on, it justifies those aspects in light of expert expertise. Furthermore, this study evaluates different CNNs rigorously, taking into consideration what factors were used to determine illness state, and justifies their clinical utility.
- Exploiting the advantages of CNN and ResNet with images, this work utilizes ResNet-101, ResNet-50, and VggNet-16 to recognize the slight differences and classify DR images.
- This study also compares these three networks to determine the best for this kind of image.

The rest of this paper is organized as follows. Section 2 introduces the related work of this study, while Section 3 highlights the suggested methods. Section 4 establishes the experiments and highlights the performance evaluation of the CNN architectures, and Section 5 finally concludes this work.

## 2. Related Work

The primary research on DR detection essentially divides DR problems into different components, such as vascular, microaneurysm, fovea, exudate, hemorrhage, and optic disc, to analyze the severity of DR. In [15], a multidirectional two-dimensional Gaussian-matched filter is proposed to detect blood vessels. In [16], sinthanayothin attempts are made to localize the disc by discovering the area with the biggest local average gray variance. Baudoin finds microaneurysms in fluoroscopic photographs of the fundus according to morphological methods in [17]. Although these methods can locate the components of the retina well, they usually operate on high-quality images and cannot directly detect the severity of the DR.

For DR harshness classification, many traditional machine learning methods are used to detect DR automatically. Most traditional machine learning methods first extract features by hand, then classify images through traditional classifiers. In [18], a model of different types of retinopathy is developed based on scale-invariant feature transform in [19,20] to generate an image local descriptor. In [21,22], the BoVW scheme is based on the detection of low-level sparse features with an accelerated enhanced features (i.e., SURF) local descriptor and midlevel features in semisoft encoding with the maximum combination. The best representation of BoVW for retinal image classification is area under the receiver operating characteristics curves of 97.8% (exudates) and 93.5% (red lesions), applying a set of data cross-validation protocols. In [23,24], Seoud first generates the injury probability map and then combines its position, size, and probability information to represent the characteristics. The proposed approach achieves an overall FROC curve score of 0.393, while a previous work with images of similar resolution reports a score of 0.233. In [25], by extracting the features of the vascular and exudate areas and the texture features, a support vector machine (SVM) is provided for DR detection using fundus images. In [26], a random forest is used as a basic classifier. In [27], SVM is used; although the SVM and random forest methods can detect DR automatically, they are sensitive to the quality of fundus images and are inappropriate for large-scale datasets. In [28,29], the optic disc is detected using morphological filtering and watershed transformation techniques, The algorithm is tested on a small image database, obtaining a mean sensitivity of 92.8% and a mean predictive value of 92.4%, and compared with a human evaluator in terms of performance.

In recent years, deep learning technology has brought about an extraordinary revolution in the field of computer vision. The use of CNN for image classification has especially attracted the attention of many investigators in this area, including the segmentation of these features and blood vessels, as mentioned in [30,31]. The deep CNN structure was initially proposed to solve the problem of natural image classification. In [32], a CNN model (LeNet-5) is used to extract image features to solve the vascular segmentation problem. These technologies have some limitations. First, given the manual extraction of dataset features by experts, their accuracy cannot be ensured. Second, the dataset is limited and of poor quality, usually consisting of just a few hundred or even hundreds of fundus images, and the acquisition setting is rather simple, making it impossible to compare the algorithms' performance in the experiment. Since Alex et al. [33,34] won in the 2012 ILSVRC competition, a remarkable improvement in the AlexNet architecture has been proposed, and the CNN's deep-field computer vision has been extensively applied. Several excellent CNN architectures have also been recommended to further improve CNN's performance, such as VggNet [35,36], GoogleNet [37], and ResNet [38]; the latest is one of the most important network classification models proposed in 2015. CNN models (AlexNet, VggNet, GoogleNet, and ResNet) are employed in [39] to distinguish the minor changes across image classes for DR detection, with the best classification accuracy of 95.68%. In [40], two models of deep learning are adopted. The first model (CNN512) uses the full image as the input to the CNN model for classification in one of the five DR classes, obtaining 84.1% and 88.6% accuracies on the APTOS and DDR Kaggle 2019 public datasets, respectively, and the second model uses an adopted YOLOv3 model to detect and locate DR lesions, reaching 0.216 mAP in locating lesions in all of the DR detection data.

According to [41], the SVM is suggested to predict DR in three separate classifications, including diabetic normal, nonproliferative retinopathy (non-PDR), and diabetic PDR, based on the features contained in a picture of the entry retinal fundus. In addition, the CNN technique can be applied to DR images. In [42], a four-layer DCNN is used to divide DR into normal, mild, and severe DR. In [43], a universal representational model, which enables the effective CNN classification of target objects inside images at any scale, is presented. The efficacy of this method is proven in a real-world application of detecting lesions in retinal pictures for classes that differ significantly in scale. Recently, Ref. [44] used a CNN based on Vgg16 on the Kaggle DR database [45] and achieved a sensitivity of 0.95 and an accuracy of 0.75 on 5000 verification images. These techniques essentially focus

on classification without the capability to locate distinctive areas. Ref. [46] proposes class activation mapping (CAM), which is used to describe the weighted activation mapping after the global average pool [47], to locate different image areas. Inspired by CAM, [30] extended the method [35] from classification to a regression locale and shed light on the DR detection difficulty, which is based on VGG16 without fully connected layers.

According to Ref. [48], an embedded system based on image processing might be used to identify conditions that lead to blindness, such as glaucoma and diabetic retinopathy. For low-cost automated glaucoma and retinopathy diagnosis, a hybrid feature extraction technique is described. The datasets are classified using an artificial neural network classifier. In Ref. [49], initially, the RGB retina image is transformed to grayscale. On this grayscale image, contrast-limited adaptive histogram equalization (CLAHE) is used to adapt the varied intensity variances to a uniform intensity. Then, to reduce background noise and improve blood arteries, morphological opening surgery is performed. Later, using canny edge detection, the perimeter is recovered from the morphing image. After that, gray thresholding is used to remove the region from the morphing image. The retinopathy is then visible in the resulting image.

A methodical investigation of the significance of image processing for diabetic eye disease (DED) categorization is presented in Ref. [50]. Picture quality enhancement, image segmentation (area of interest), image augmentation (geometric transformation), and classification are all steps in the proposed automated classification framework for DED. Traditional image processing methods are combined with a novel built convolution neural network (CNN) architecture to produce the best results. For DED classification tasks, the novel designed CNN paired with the old image processing approach provides the best result in terms of accuracy. In [51], the input image is collected from the Indian Diabetic Retinopathy Image Dataset database, and 13 filters are used to enhance the photos, including smoothing and sharpening filters. Then, using performance measures, the quality of the enhancement algorithms is compared, and superior results are obtained for the Median, Gaussian, Bilateral, Wiener, and partial differential equation filters, which are integrated to improve picture enhancement. The convolutional neural network input is supplied to all of the enhanced filters' output images, and the results are compared to discover the best enhancement approach.

Researchers working on the retinopathy detection problem ran into a few issues, many of which are unavoidable and for which there is no immediate solution. This is because the field of deep learning is still relatively new to many people, and data collection is always problematic. Data are sparse, and much of what is accessible is unusable for one reason or another. Using CNN to separate the lesions, Figure 2 shows the research gaps in retinopathy. The following are the key drawbacks of traditional approaches and DL architectures, as determined by the preceding study of the existing literature: 1. Dataset is limited, 2. Images that are twisted and blurred, 3. Models that are overfitting, and 4. Limited computing power.

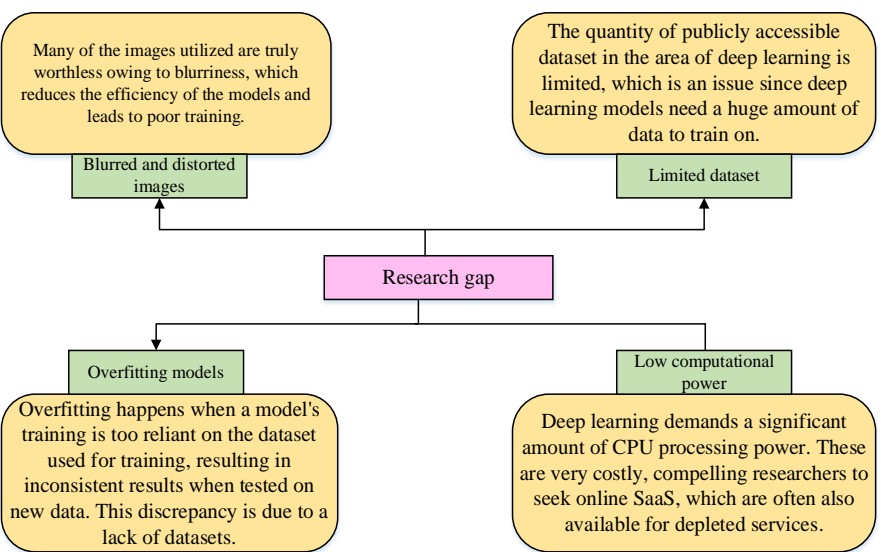

**Figure 2.** Study of the research gap of Retinopathy using CNN to segment the lesions.

## 3. Methods

First, some augmenting and preprocessing methods were used to increase the amount of training data and improve the classification effect and performance of the deep learning network used in the laboratory dataset. For example, the input images were resized and augmented, and the data was normalized. Then, the three most advanced CNN models were used to detect DR, and the accuracy of these models in image classification was tested.

### 3.1. Proposed Solution for Solving the Dataset Issues

3.1.1. Data Preprocessing

Given that low-quality images will produce inaccurate results, preprocessing is an essential operation to improve image quality. The results of preprocessing are regarded as the original input for data training, which divides the images into two categories. The dataset also contains images from patients of different races, age groups, and lighting levels in fundus photography, which will affect the pixel intensity values in the image and produce unnecessary changes independent of the classification level. To solve this problem, this work used the OpenCV library to preprocess the images. The OpenCV library provides two transform functions, cv2.warpAffine and cv2.warpPerspective, and noise. This library is necessary for image processing, as it comprises rich built-in features for quick processing.

The preprocessing consisted of image cropping followed by resizing. Each image was cropped to a square shape, which included the most tightly contained circular area of the fundus. The procedure removed most of the black borders and all of the patient-related annotations from the image data. Each of the cropped images was then resized to 300 × 300 pixels. Here, the creation of multiple resolutions was for the purposes of analyzing the effect of the input image resolution on the classification performance

- Original fundus image dataset and retinal image grading systems

The research of the present study was conducted in collaboration with Xiangya No. 2 Hospital Ophthalmology (XHO), Changsha, China, a certified provider of diabetic retinopathy screening and monitoring services in China (obtained from the Xiangya Second Ophthalmic Hospital, Changsha, China, from March 2016 to October 2016). The output images were of variable resolutions of 1956 × 1934 pixels.

Each of the retinal images was graded with respect to three different criteria: (i) diabetic retinopathy, (ii) macular edema, and (iii) gradability, and a number of images were not graded to any criteria. Images are graded with the proposed international clinical diabetic retinopathy and macular edema disease severity scales, denoted later as symptomatic and asymptomatic, respectively. Image gradability is a two-stage system, which considers an

image to be either gradable or not. All personnel participating in retinopathy assessment had over 10 years' experience in diabetic retinopathy grading. All images with symptomatic and asymptomatic were graded by two ophthalmologists, both with more than 10 years of experience in grading fundus images. If there was a disagreement in grading, such an image was not included in this study.

Finally, all the images classified into (symptoms 607 and nonsymptoms 1000) images from the training set were used to develop and validate the model. The model was tested on (symptoms 200 and nonsymptoms 122), images as explained in Table 1.

**Table 1.** Classification dataset.

| Class Name | Degree of DR | Number of Labels | For Training | For Testing |
|:---:|:---:|:---:|:---:|:---:|
| Class 0 | Nonsymptoms | 1000 | 800 | 200 |
| Class 1 | Symptoms | 607 | 485 | 122 |

Images from the training dataset (samples in Figure 3) are read using the Tensorflow-GPU library. This library is important for image processing, as it contains rich built-in features for rapid processing. The result of this can be seen in Figure 4b; therefore, preprocessing is the reshaping of input data. The images acquired are of size 1956 × 1934 pixels and occupy a large area of memory space. This demands too much RAM usage, resulting in slower computation. Hence, they are downscaled to 300 × 300 dimensions. The result of an image augmentation can be seen in Figure 4c.

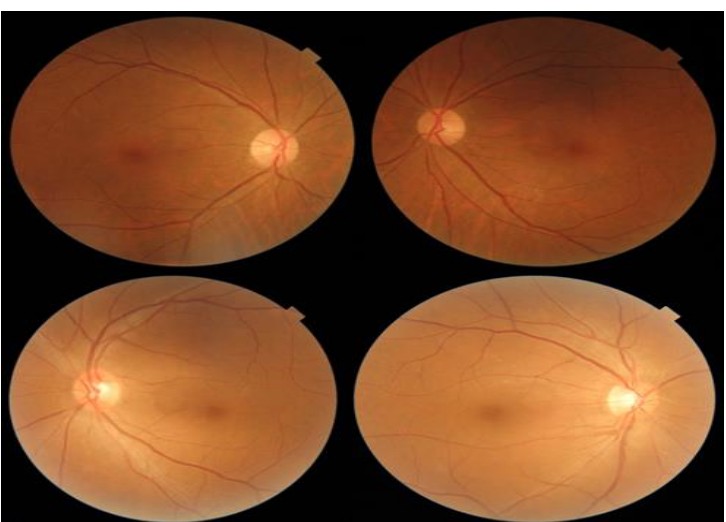

**Figure 3.** Sample frames of the retina images. The first two frames in the top row come from normal subjects, while the two frames in the bottom row come from the patients who have diabetic retinopathy.

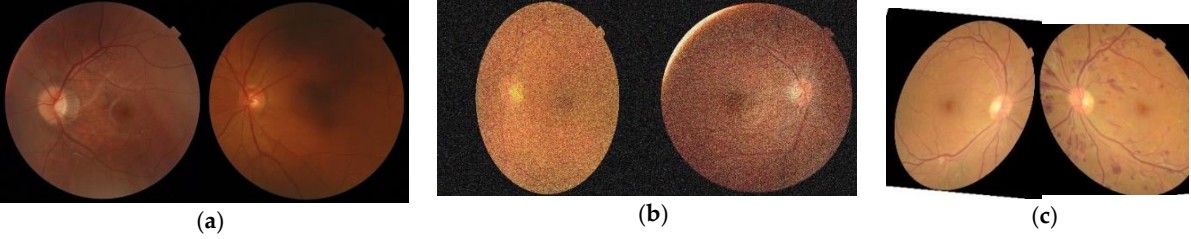

(**a**)  (**b**)  (**c**)

**Figure 4.** Illustration of the preprocessing and augmentation processes. (**a**) Original images. (**b**) Preprocessed images. (**c**) Augmented images.

### 3.1.2. Data Augmentation

The deep learning (DL) model has always been a data requirement due to data noise and dataset limitations. Thus, the data expander is used to generate more data without actually collecting more data to expand the dataset. This operation can be performed using the Keras library's built-in capabilities. Image rotation (to induce rotation invariance), horizontal flipping, scaling, clipping, and translation for preprocessing are some of the augmentation techniques. To generate new data, a recent work [52] used the most advanced generative adversarial networks (i.e., GAN). Additionally, to eliminate extraneous parts, the fundus photos are cropped to a lesser size.

### 3.1.3. Utilizing CNN Models

This paper is focused on the use of CNNs in the diagnosis of DR. Automatic feature extraction and efficient computation are hallmarks of CNN and any deep learning paradigm.

The convolution, pool, and activation layers make up the CNN model. Different filters combine the data fed into the network. These filters are similar to traditional image processing filters. However, instead of being explicitly defined, they are self-learning. The convolution layer extracts local features of various positions from the original input or intermediate feature maps with various kernel sizes. The weight-sharing mechanism, local connectivity, and target position invariance are the advantages of the convolution layer. The pooling layer shrinks the feature maps and network parameters while keeping critical information. The most commonly used methods are average pool and maximum pool. The convolution layer is usually followed by the pooling layer. The fully connected layer resembles a traditional neural network and is added to the final pool layer for classification [52].

- VggNet-16

VggNet-16 is a CNN structure, which won the 2014 ILSVRC (Imagenet) competition. VggNet-16 is regarded as one of the most advanced visualization model architectures to date. The 16 in VggNet-16 means that it has 16 layers of weight. This network is a fairly large network with approximately 138 million parameters. Figure 5c shows the VggNet-16 network structure, the number of parameters for each level, and the detailed performance test for all layers of the network [35,37,39].

VGG16 is a proposed model that compresses the previously successful VGG16 network and improves on the following aspects: (1) smaller model size, (2) quicker speed, (3) leverages residual learning for faster convergence, better generalization, and degradation resolution, and (4) equals the recognition accuracy of the noncompressed model on the very large-scale grand challenge MIT Places 365-Standard image dataset. The suggested model is 88.4% less in size and 23.86% faster in training time than VGG16.

This backs up our contention that the proposed model takes the best features of VGG16 and improves on them.

This work trains deep models of VGGNet-16 to screen DR as deep models achieve great success in many tasks. VGGNet exhibits excellent performance but is not better than ResNet-101 and ResNet-50.

- ResNet (50 and 101)

Ref. [34] proposes ResNet, which was utilized to successfully train 152 deep neural networks to win the ILSVRC 2015 championship and obtain a 3.57% error rate classification for the top 5 classes using the residual unit, which is extremely good considering the number of parameters is smaller than that in VggNet. HighWay Nets, the ResNet core, uses the skip connection to let some input into (skip) the layer to integrate the information flow indiscriminately, preventing information loss and gradient vanishing (which also suppresses the generation of some noise). Furthermore, noise suppression entails averaging the model, ensuring that the model maintains a balance in training accuracy and generalization. Ultra-deep neural network training and model accuracy can be greatly improved using the ResNet structure. ResNet solves the degradation problem by increasing the CNN depth. The accuracy initially

increases, reaches the limit, and then decreases as the depth increases. This phenomenon is not an overfitting issue because the error increases not only in the test samples but also in the training samples. When a shallow medium network satisfies the saturation accuracy and contains several congruence mappings layers, the errors will not accumulate when the network is at its smallest, and the deeper the network is, the fewer the training example errors will be. ResNet was inspired by the idea of employing a congruent mapping to send the previous output directly to the next layer. If a CNN's input is x and the expected output is H(x), our learning goal is F(x) = H(x) − x when input x is directly transferred to the output.

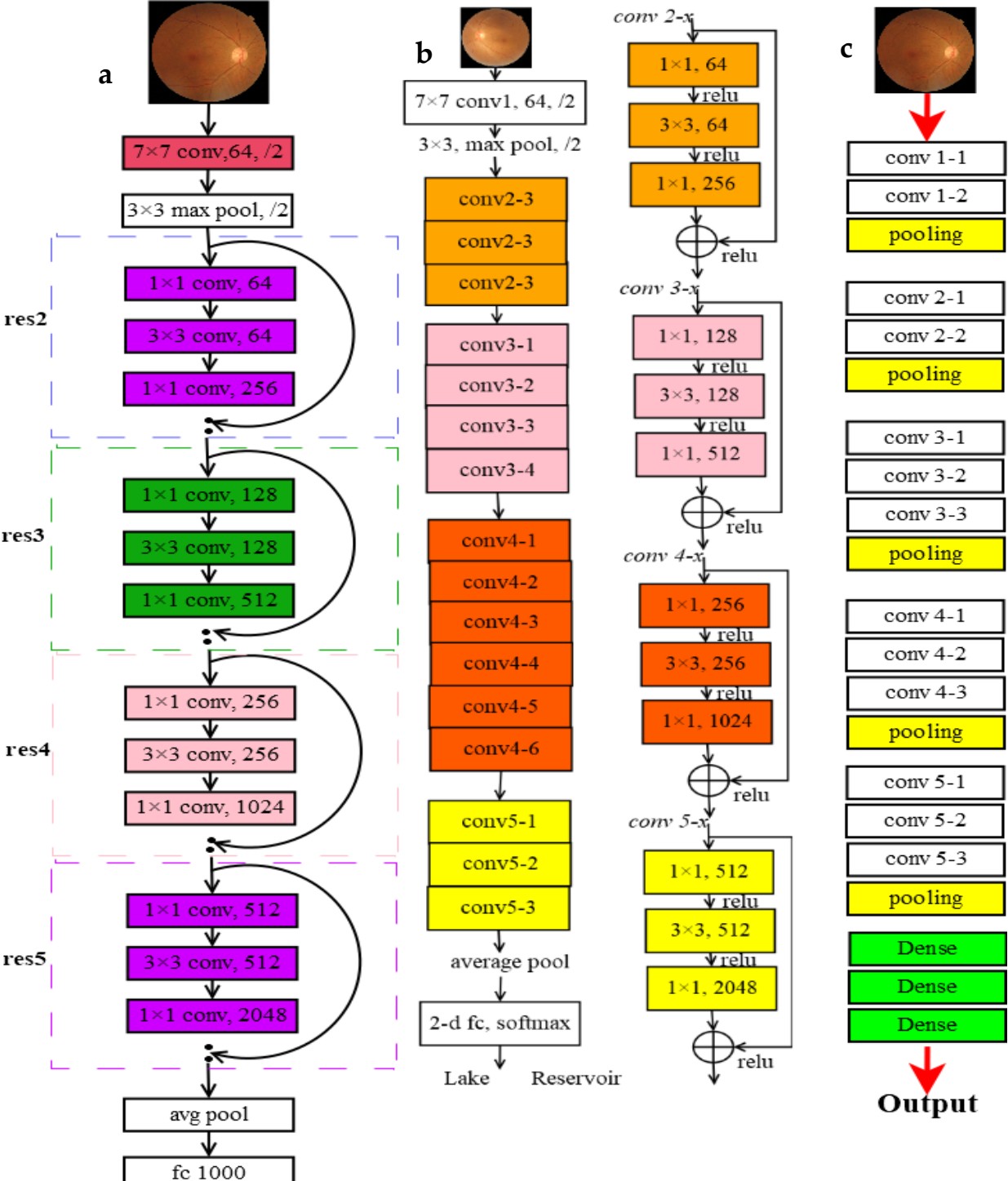

**Figure 5.** Latest CNN (**a**) ResNet-101, (**b**) ResNet-V1-50, and (**c**) VGG-16 Architecture [39].

A deep convolutional neural network with 50 layers is called ResNet-50. The ImageNet database contains a pretrained version of the network that can be loaded. This network has been trained on more than a million images. As a result, the network has acquired knowledge about several rich feature representations for numerous photos. ResNet-101 is a deep convolutional neural network with 101 layers. The ImageNet database, which has been trained on more than a million images, allows you to import a pretrained version of the network. In order to classify photos into 1000 different object categories, the network has already been trained. The network has learnt a range of rich feature representations for a variety of pictures as a consequence.

This study trained deep models VggNet-16, ResNet-50, and ResNet-101 to screen DR as deep models achieve great success in many tasks. For example, these methods have been cited for their good performance in image classification [39]. Figure 5a–c shows the ResNet-101, ResNet-50, and VggNet-16, respectively, network structures used to train our lab dataset for classifying DR images. ResNet-101 provides better results than ResNet-50 and VggNet-16.

### 3.1.4. Training Process

This work chose the latest CNN models (ResNet-101, ResNet-V1-50, and VGG-16) to train and classify our lab dataset because they are very popular for their effectiveness and robustness. They are designed in such a way that the input retinal image of size $300 \times 300$ can be fed into the network, which has alternate convolution and pooling layers activated using the ReLU activation function. The latest CNN models have been implemented in Python using Tensorflow. The parameters of the hyperparameter-tuning method are not initialized by the network itself; it is necessary to tune and optimize these parameters according to the results of training the DR image in enhancing performance. In addition, some of these parameters used in the proposed methods are mentioned in Table 2.

**Table 2.** Parameter tuning in the latest CNN models.

| Description | Output Shape |
|:---:|:---:|
| weight_decay | 0.00001 |
| num_classes | 2 |
| batch_size | 8 |
| val_batch_size | 32 |
| image_height | 300 |
| image_width | 300 |
| image_channels | 3 |
| num_iters | 4000 |
| lr | [0.0001, 0.001] |
| momentum | 0.9 |
| batch_norm_scale | True |
| batch_norm_epsilon | $1 \times 10^{-5}$ |
| batch_norm_decay | 0.997 |

In this work, 1607 images from the Xiangya No. 2 Hospital Ophthalmology Changsha China dataset and 1000 images from this dataset are healthy and 607 images are defective. Out of 1000 healthy images, 800 images are used for training, and the remaining 200 images are used for testing; out of the 607 defective images, 485 images are used for training and 122 for testing. The images are fed into the network as tensors, and the network is trained iteratively. The hyperparameters of the proposed models are fine-tuned to achieve better performance. The number of iterations for every retinal image is set as 4000, and the learning rate is chosen as 0.0005. Parameter tuning for the proposed models is given in Table 2.

## 4. Experiment

This work started by using some preprocessing methods. The input image was resized and augmented and the data normalized to increase the amount of training data and improve the performance of using a deep training network for this work's laboratory dataset. Then, the three state-of-the-art CNN models were used to detect DR, and the accuracy of these models was tested to classify the images. In addition to measuring the number of correctly and incorrectly classified test images, the images must also be evaluated according to the categories of error classification and discipline the accuracy scores accordingly.

### 4.1. Dataset and Preprocessing

The images were obtained from Xiangya No. 2 Hospital Ophthalmology (XHO), Changsha, China, for the study of two causes or classes of diabetic patients with and without symptoms. Initially, the datasets were indivisible, unmodified, and dirty due to many datasets covering many diseases, which are not diabetes. We then modified, arranged, and divided them into two appropriate categories of diabetes with and without symptoms. Second, all the images used were preprocessed to avoid noise and poor illumination. Before training the collected dataset, this study used many preprocessing methods to prepare the input images. The preprocessing included cropping and scaling the images. Each image was cropped to a square form, which includes the circular section of the fun-dust that is most securely enclosed. The images acquired are of size $1956 \times 1934$ pixels and occupy a large space. This requires higher RAM usage, which leads to low computation. Hence, they are downscaled to $300 \times 300$ dimensions before feeding them into the framework. This helps speed up the performance.

The number of images used with symptoms is 607, while 1000 did not have symptoms. Given the insufficient number of images for training, this study increased the size of the dataset by using the augmentation steps as shown in Table 3. Furthermore, given the presence of noise and the limited number of datasets, increasing the data is always preferred. Therefore, a data expander was used to generate more data without actually collecting more data. This work used two different types of OpenCV methods to increase the training set by rotating images and providing noisy images to improve model accuracy, as shown in Figure 6. The XHO dataset was divided into 800 for the no symptoms class and 485 for the symptoms class in the training dataset before the augmentation operation. The images in the testing dataset are 322. After the augmentation, the datasets increased to 1982 for the no symptoms class and 1204 for the symptoms class in the training dataset. The initial learning rate was 0.001, the momentum was 0.9, the batch size was 800, and the iterations were 3000.

**Table 3.** The number of images before and after the augmentation process.

| Class | All | Training Set | | Tasting Set |
|---|---|---|---|---|
| | | Before Augmentation | After Augmentation | |
| No symptoms | 1000 | 800 | 1982 | 200 |
| Symptoms | 607 | 485 | 1204 | 122 |

### 4.2. Results and Discussion

The first experiment was on 16 layers of the VGG network for classifying the input images. The different scales of the 16 CNN layers and the max-pooling layers after each different scale were used to avoid overfitting and speed the processing operation. VGG-16 successfully obtained the training and testing classes, as shown in Table 4. Meanwhile, the accuracy is not very high because of the problem of vanishing gradients, which results in an insufficient number of layers for extracting the many features of the input images.

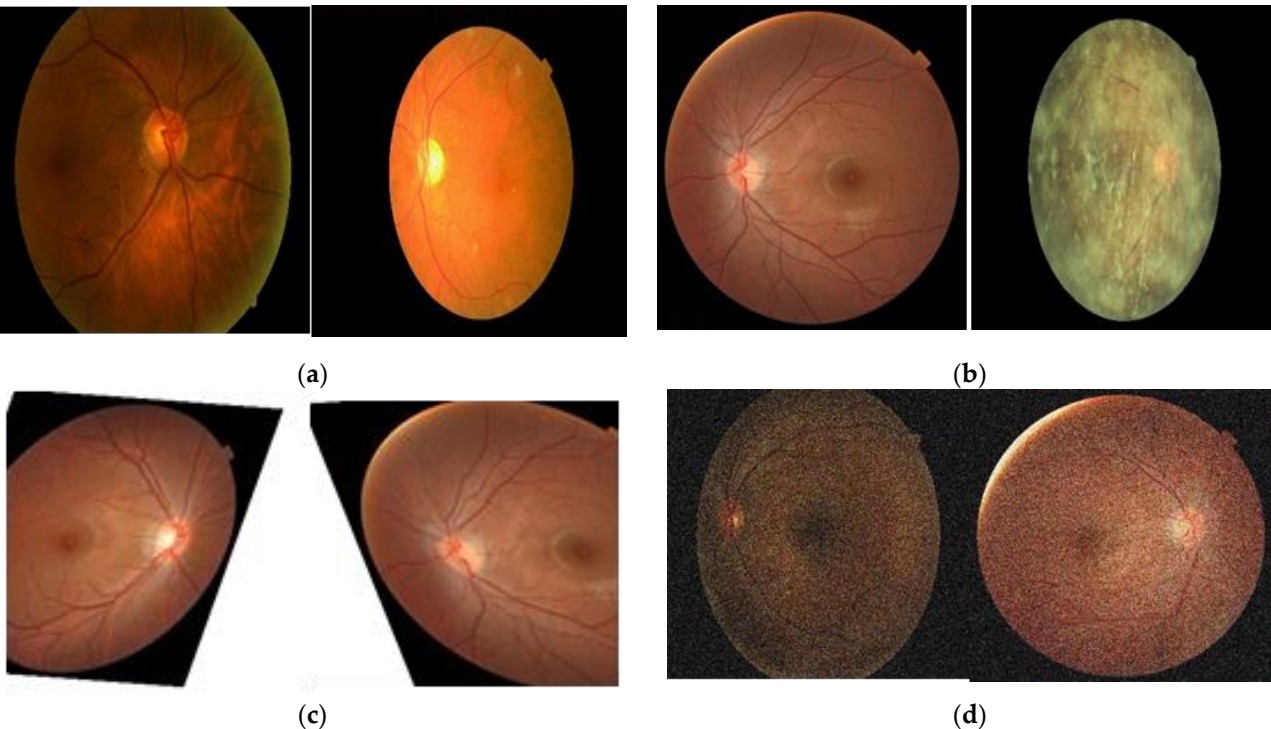

**Figure 6.** Some examples of images in the training set and the output of preprocessing to increase the size of the input dataset. (**a**) Images have symptoms. (**b**) Images without any symptoms. (**c**) Images with left and right rotate. (**d**) Images with noise.

**Table 4.** Performance comparison with different approaches before augmentation.

| Model | Training Accuracy (%) | Training Loss (%) | Testing Accuracy (%) |
|---|---|---|---|
| ResNet-101 | 80.88 | 40.1 | 79.76 |
| ResNet-50 | 79.7 | 37.6 | 71.8 |
| VggNet-16 | 64.4 | 67.32 | 62.11 |

The second experiment used ResNet-50. This network contains different scales of 50 CNN layers with residual blocks to avoid the vanishing gradients problem. As shown in Figure 5, each block has a particular number of CNN layers and scale, and the input of each block is added to the output of the same block, helping enrich and increase the feature extraction. The accuracy of this experiment is higher than that of VGG-16 because of the ResNet properties.

To improve the model accuracy, we increased the number of layers in Resnet to 101 in the last experiment. The advantage of ResNet-101 is that it has more layers, is easier to overfit, and has a deeper network structure than ResNet-50. ResNet-101 has 101 layers, helping obtain more features of the input images. ResNet-101 achieved the highest accuracy in terms of classifying the input images for both the training and testing datasets, as shown in Tables 4 and 5. The cross-entropy loss function was used to evaluate the training operation in all experiments to adjust the model weights during training. In addition, we used the arithmetic average to assess the model accuracy when the model was tested with the testing dataset.

The learning performance of the CNN models (ResNet-101, ResNet-50, and VGG-16) are shown graphically in Figure 7a,b, Figure 8a,b and Figure 9a,b, respectively. These figures show the plot of the number of iterations on the *x*-axis against the accuracy and loss on the *y*-axis. The ResNet-101 model exhibited promising results and robust stability in classifying the images as either healthy or afflicted by scanning the existence of DR.

The designed model proves efficient in classifying the images present in the XHO datasets because the ResNet-101 model has a higher accuracy than ResNet-50 and VggNet-16.

**Table 5.** Performance comparison with different approaches after augmentation.

| Model | Training Accuracy (%) | Training Loss (%) | Testing Accuracy (%) |
| --- | --- | --- | --- |
| ResNet-101 | 98.88 | 34.99 | 98.82 |
| ResNet-50 | 93 | 34 | 91.5 |
| VggNet-16 | 71.39 | 61.48 | 64.11 |

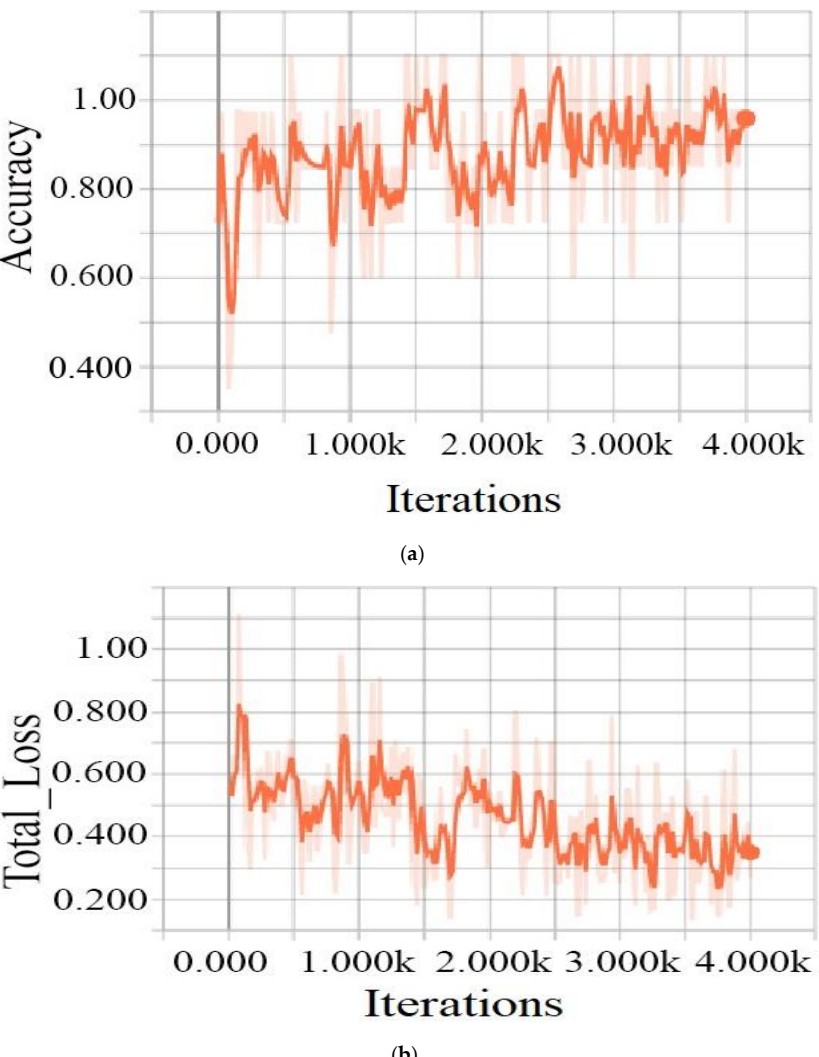

(**a**)

(**b**)

**Figure 7.** Training process of ResNet-101 for both (**a**) accuracy and (**b**) loss.

The classification accuracies are shown in Tables 4 and 5. The tables indicate that the results obtained using the CNN models with data augmentation are better than those obtained without data augmentation because data augmentation can assist CNN models in dealing with minor rotations or translations during data recording. Additionally, because of the imbalanced distribution of the XHO dataset in which the number of normal images was more than that of the DR images, we first classified the images into two DR stages and split them into the training and testing data sets. Then, image preprocessing was performed to improve the quality of the retinal images, which is important because low-quality images can degrade network performance. Thus, ensuring the consistency of all images and the enhancement of the features of the images are important. Thus, as networks and datasets

improve and real-time classifications become available, the potential usefulness of CNN models (ResNet-101, ResNet-50, and VGG-16) to DR clinicians will continue to increase.

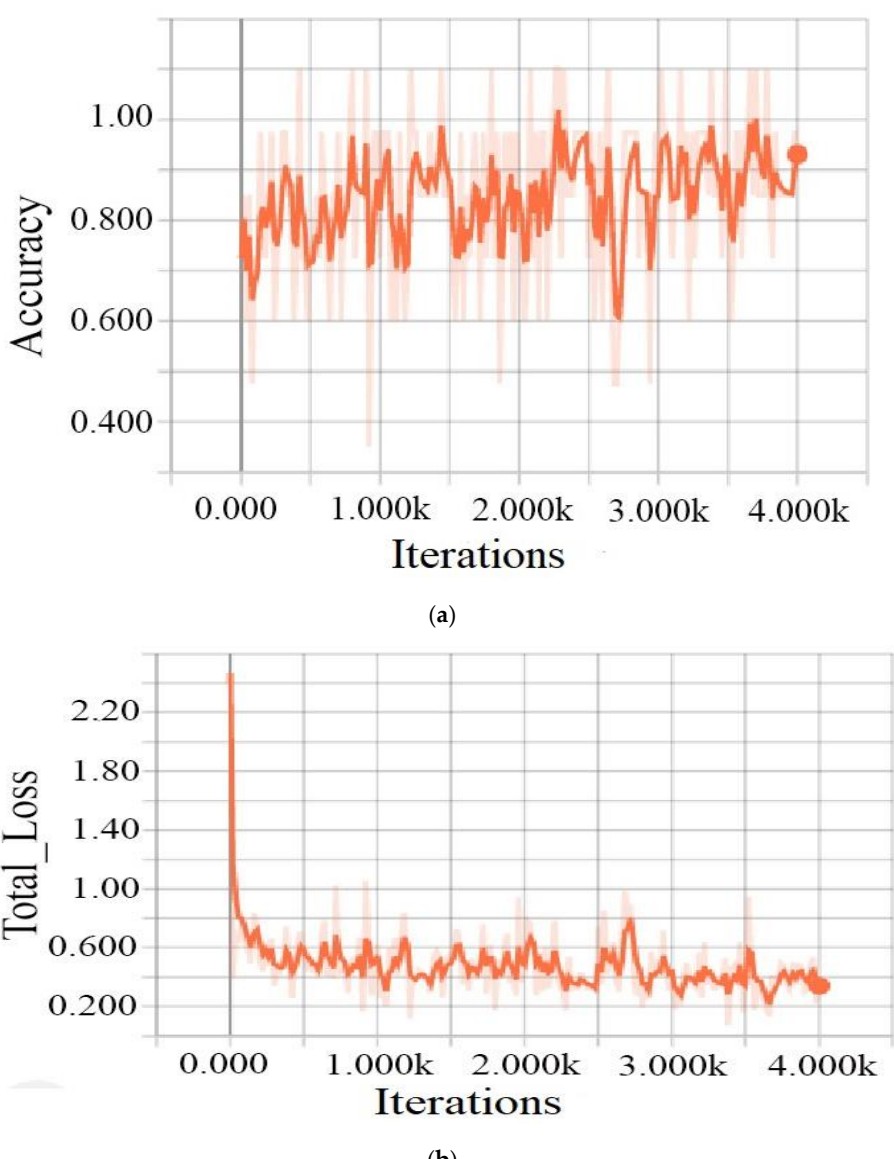

**Figure 8.** Training process of ResNet-50 for both (**a**) accuracy and (**b**) loss.

### 4.3. Performance Evaluation

To assess the performance of the proposed system ResNet-101 in DR detection on our dataset and other datasets, we used a high-resolution fundus (HRF) image, which contains 30 images of dimension 3304 × 2336 pixels [53]; STructured Analysis of the Retina (STARE), which contains 20 images from the STARE dataset of size 700 × 605 pixels [54]; DIARETDB0, which contains 130 images from the DIARETDB0 dataset of size 1500 × 1152 pixels [55]; 1200 images from the MESSIDOR dataset of resolution 1440 × 960 [56]; and our database XHO, which contains 1607 images of size 300 × 300.

The testing dataset is divided into two categories: images with no DR and images with DR recognized by ophthalmologists who divided the 2987 images with DR into 1089 images. No DR was observed when the suggested approach was tested on normal (no DR) pictures. Table 6 shows a detailed summary of all datasets used.

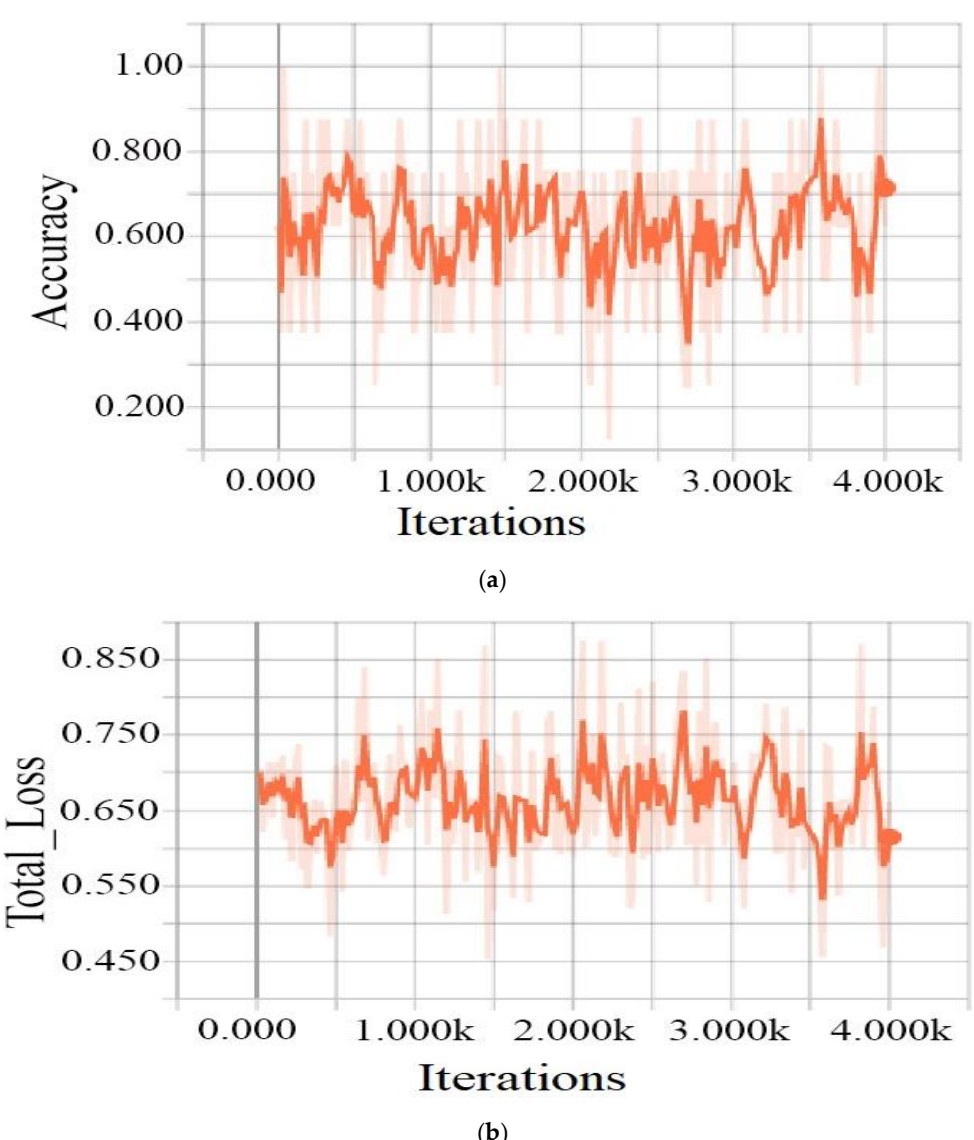

(**a**)

(**b**)

**Figure 9.** Training process of VggNet-16 for both (**a**) accuracy and (**b**) loss.

**Table 6.** Complete description of database.

| Dataset | Number of Images | No DR | DR |
|---|---|---|---|
| XHO datasets | 1607 | 1000 | 607 |
| HRF datasets | 30 | 15 | 15 |
| STARE datasets | 20 | 12 | 8 |
| DIARETDB0 datasets | 130 | 20 | 110 |
| MESSIDOR | 1200 | 851 | 349 |
| Total | 2987 | 1898 | 1089 |

*4.4. Performance Evaluation Criteria*

The measures used to evaluate the effectiveness of CNNs include accuracy (ACC), specificity (SP), sensitivity (SEN), area under the ROC curve (AUC), positive predictive value (PPV), negative predictive value (NPV), and F1 score (F1). The amount of correctly identified photos is known as the ACC. The SP is the proportion of correctly categorized photos as normal images, while the SEN is the proportion of correctly classified images as DR images. The ROC curve and the value calculated by ROC, and the AUC provide a graphic representation of the ratio between SEN and SP. PPV is the proportion of correctly categorized DR pictures, whereas NPV is the proportion of correctly labeled normal images.

These variables are used to evaluate how ResNet-101 performs in comparison to other algorithms. The following defines these metrics:

$$SP = \frac{TN}{(TN + FP)} \tag{1}$$

$$SEN = \frac{TP}{(TP + FN)} \tag{2}$$

$$ACC = \frac{(TN + TP)}{(TN + TP + FN + FP)} \tag{3}$$

$$PPV = \frac{TP}{(TP + FP)} \tag{4}$$

$$NPV = \frac{TN}{(TN + FN)} \tag{5}$$

$$F1score = 2 * \frac{Precision * Recall}{Precision + Recall'} \tag{6}$$

where false negative (FN) values refer to DR pictures that are categorized as non-DR, whereas false positive (FP) values refer to non-DR images that are labeled as DR. The terms TP and TN stand for true positive and true negative values, respectively, and relate to the DR pictures that are categorized as DR and the non-DR images that are not.

ResNet-101 is implemented utilizing many assessment criteria, including ACC, SEN, SP, and F1. The usage of 2987 retinal fundus pictures from four distinct datasets. For the detection of DR, an assessment of the proposed system ResNet-101 is conducted. The results of DR detection using various datasets were shown in Table 7 for all the accuracy (ACC), specificity (SP), sensitivity (SEN), area under the ROC curve (AUC), and F1 score metrics (F1). The AUC values are evaluated for the HRF, DRIVE, STARE, MESSIDOR, DIARETDB0, and DIARETDB1 datasets. The ROC curve plot is shown in Figure 10. In Table 8, the findings are compared with several current state-of-the-art methods for each dataset to determine their superiority and efficacy.

**Table 7.** Summary of DR detection for four different datasets using ResNet-101 CNN method.

| Dataset | Test Images | Correctly Detected | Accuracy (%) | Sensitivity (%) | Specificity (%) | F1 Score (%) | AUC (%) |
|---|---|---|---|---|---|---|---|
| XHO datasets | 200 | 196 | 98 | 97.14 | 97.65 | 97.36 | 98.55 |
| HRF datasets | 30 | 30 | 100 | 99.98 | 99.98 | 99.98 | 99.99 |
| STARE datasets | 20 | 19 | 95 | 94.96 | 95.11 | 95.03 | 95.04 |
| DIARETDB0 datasets | 110 | 105 | 95.45 | 95.39 | 99.38 | 95.45 | 95.46 |
| MESSIDOR | 349 | 347 | 99.42 | 99.45 | 99.38 | 99.41 | 99.42 |
| Total | 360 | 349 | 97 | 96.87 | 98.03 | 96.95 | 97.26 |

**Table 8.** Comparison between the related works that used CNN with our work to classify DR Images.

| Evaluated Parameter | | [57] | [58] | [59] | Proposed Method | [60] | [39] | [61] | [62] | [63] |
|---|---|---|---|---|---|---|---|---|---|---|
| Number of Classes | | | | 2 | | | 5 | | 4 | 5 |
| Detect Lesion | | | | No | | | No | | No | Yes |
| Dataset | | | | private dataset | | | Kaggle | | Messidor | DDR |
| Performance | ACC | 94.23% | 88.21% | 98.7% | 98.88% | 63.23% | 95.6% | 98.15% | 96.35% | 82.84% |
| Measure | AUC | 0.9823 | 0.946 | - | 98.55% | - | 0.978 | - | - | - |

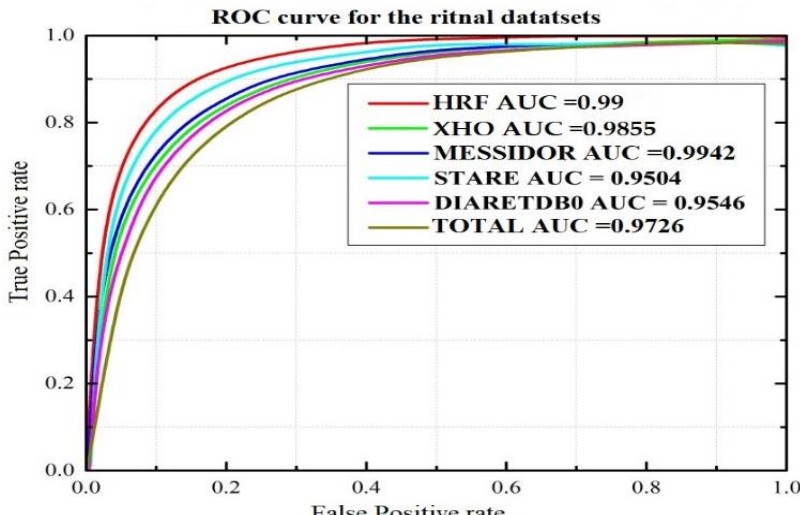

**Figure 10.** ROC plot for retinal HRF, DRIVE, STARE, MESSIDOR, DIARETDB0, and XHO datasets.

## 5. Conclusions

The automatic classification of DR fundus images can aid doctors in the diagnosis of DR and accelerate the diagnosis process. In this study, we first established the preprocessing and regularization steps that our lab dataset images require for maximizing the deep learning system's functionality. Then, we looked into how much a small number of images used in training affects the performance. Second, our work has shown that the two-class problem for national screening of DR can be approached using a CNN method. The latest CNN models (ResNet-101, ResNet-50, and VggNet-16) show promising signs of being able to learn the features required to classify fundus images. The ResNet-101 model was tested on a set of images from our lab datasets. The network achieved a testing accuracy of 98.8198758%, whereas the network provides a training accuracy of 98.88% with a training loss of 34.99%. Finally, the ResNet-101 method was applied to 1787 color fundus images from the four (HRF, STARE, DIARETDB0, MESSIDOR, and XHO) datasets, and achieved an accuracy of 98%, 100%, 95%, 95.45%, and 97% for the all datasets, respectively. Therefore, the latest CNN models (ResNet-101, ResNet-50, and VggNet-16) can be trained to identify the features of DR in fundus images.

Our data sets' quality and balance are essential for developing a system for DR detection. To create a balance for these datasets, we plan to merge several datasets in the future. Although the CNN models work well in detecting DR in two stages of our datasets, they have limitations in terms of the classification of system performance, so our goal is to extend and improve these models by adding more layers or by using CNN's new model design, which can categorize and produce results in real time.

**Author Contributions:** Conceptualization, A.-O.A. and C.-Z.Z.; methodology, M.A.A.A.-Q.; software, D.A.-A.; validation, A.-O.A., D.A.-A. and S.A.A.; formal analysis, P.-B.O.; investigation, A.-O.A.; resources, S.A.A.; data curation, Y.-L.X.; writing—original draft preparation, A.-O.A.; writing— review and editing, M.A.A.A.-Q.; visualization, C.-Z.Z.; supervision, C.-Z.Z.; project administration, A.-O.A.; funding acquisition, S.A.A. All authors have read and agreed to the published version of the manuscript.

**Funding:** This work is supported by the National Key R&D Program of China (2018AAA0102100), the Scientific and Technological Innovation Leading Plan of High-tech Industry of Hunan Province (2020GK2021), the Natural Science Foundation of Hunan Province (No. 2022JJ30762), the International Science and Technology Innovation Joint Base of Machine Vision and Medical Image Processing in Hunan Province (2021CB1013), the Key Research and Development Program of Hunan Province (2022SK2054), the 111 project under grant no. B18059.

**Data Availability Statement:** All of the data in this report come from the Xiangya No. 2 Hospital Ophthalmology (XHO), Changsha, China. In addition, it is available as requested.

**Acknowledgments:** We want to express our thanks and gratitude to all those who helped us throughout this study.

**Conflicts of Interest:** The authors declare no conflict of interest.

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
