# Peer review of "Detection of Diabetic Retinopathy in Retinal Fundus Images Using CNN Classification Models"

_electronics, doi:10.3390/electronics11172740_

Round 1

Reviewer 1 Report

In this article authors described deep learning based classification for the two categories of DR (symptomatic and asymptomatic) using different kinds of CNN architectures on an in-house dataset. I have few questions regarding the manuscript: 

  1. According to the authors, this is the first work to classify symptomatic and asymptomatic DR. Please provide inter/intra user variability for manually labeling of such dataset. Also provide some examples of these two classes.  

  1. Very few details about preprocessing methods presented in the paper, which is one of the main contributions of the work. Please give more details, such as is this process (cropping location etc.) is automatic or manually done for each scan. What is the use of cv2.warpAffine and cv2.warpPerspective ? How the parameters of these functions are chosen? How the scans are down sampled ?  

  1. Section 3.1.3. Utilizing CNN models  is redundant. All the models are quite popular in the field. You can just refer them. 

  1. Please provide more details about training process and make a separate section. What is the percentage of validation set? How the final model is chosen ? What is the hyperparameters for each training?  

  1. How the model is applied to external dataset ? What was the preprocessing steps in case of these? Was model trained again for these datsets?  

  1. If large number of external dataset available for these classifications, then why not use transfer learning ? 

  1. Also provided Class Activation Maps from the test cases for better understanding of the model

Reviewer 2 Report

Dear Authors,

This work presents a "Diabetic Retinopathy" detection procedure using a pre-trained CNN scheme. This work is good and the results achieved on the clinical data are also fine.

Major suggestion:

The data availability statement "All of the data in this report come from the Xiangya No. 2 Hospital 498 Ophthalmology (XHO), Changsha, China. In addition, it is available for requested" confirms that this work considered the clinical data. If possible, include the Ethical Clearance Statement/number to consider the clinical data. Else, if this data is already used in earlier work. Please refer to and indicate the same in this paper. Further, clear information about the database and procedure followed during the examination can be presented briefly under the XHO dataset sub-section.

General Suggestion:

1.  I request you to replace, I, We, Our with an appropriate word. Further in the introduction section, please include;

(i) (a) Major contributions of this paper and (b) Organisation of the work at the end of the introduction section

(ii) Figure 2 available after Table 1 must be removed completely.

(iii) I also request you to remove Figure 3 and Figure 5 (since the information is known/not necessary in this work)

(iv) Figure 4 caption needs to be checked and corrected (Since (c) is missing)

(v) Figure 7 to 9, the epoch (X-axis) is mentioned as 4.000K. 4000 epochs are very high.  Please justify. Further, the training accuracy and loss functions have still not reached saturation. Please justify. 

(vi) Eqn. (6) must be F1 score instead "F1". Also unused abbreviations, SP, SEN can be avoided.

(vii) Graphical representation for  Table 5 can be added. 

(viii) Limitations and future scope of this work can be included.

Round 2

Reviewer 1 Report

Authors have answered all of my the questions.